# Relativistic Symmetries and Hamiltonian Formalism

**Piotr Kosiński** * and **Paweł Maślanka** *

Faculty of Physics and Applied Informatics, University of Lodz, Narutowicza 68, 90-136 Lodz, Poland
* Correspondence: piotr.kosinski@uni.lodz.pl (P.K.); pawel.maslanka@uni.lodz.pl (P.M.)

**Abstract:** The relativistic (Poincaré and conformal) symmetries of classical elementary systems are briefly discussed and reviewed. The main framework is provided by the Hamiltonian formalism for dynamical systems exhibiting symmetry described by a given Lie group. The construction of phase space and canonical variables is given using the tools from the coadjoint orbits method. It is indicated how the "exotic" Lorentz transformation properties for particle coordinates can be derived; they are shown to be the natural consequence of the formalism.

**Keywords:** coadjoint orbits; conformal group; Poincaré group

## 1. Introduction

We present here a brief review of some results obtained by our colleagues in collaboration with Yves Brihaye and us. They concern the old topic of the basic role of space-time symmetries in physics. It appears that some of the main results, which were for the first time obtained and developed in quantum theory, can be described quite precisely on the classical level in the framework of Hamiltonian formalism. Then, their quantum counterparts are recovered by applying a straightforward canonical quantization procedure. The problem is not only academic. For example, in recent years, much effort has been devoted to exploring the anomaly-related phenomena in kinetic theory [1,2]. This work is to a great extent based on semiclassical approximation and the description of various symmetries within this approximation. The sound knowledge concerning the (semi)classical aspects of relativistic symmetries contributes to a deeper understanding of such phenomena.

Part of the results reported here has been obtained in collaboration with our longtime friend Yves Brihaye. It was always a great pleasure to work with him.

## 2. Orbit Method

In physics, beauty is often identified with symmetry. Having at our disposal two theories explaining the same set of experimental data, we are inclined to choose the one exhibiting more symmetry. This strategy is somehow supported by our experience. A dazzling example is the success of general relativity, which is confirmed over and over again by experiments, observations and even everyday life. Although some physicists warn that beauty can lead one astray [3], we are, generally speaking, attached to the idea that adopting symmetry as a guiding principle often leads to the theories with considerable predictive power.

The symmetry of the physical system is described, at the formal level, by the choice of symmetry group $G$. In quantum theory, the states of the physical system are classified according to the unitary representations of $G$. However, even if the group $G$ represents the maximal symmetry of some system, there is still much freedom in the choice of the total space of states and relevant observables. The exception is provided by the so-called elementary systems, which are, by definition, described by irreducible representations of the symmetry group. For the elementary system, given the symmetry group $G$, one can classify all admissible spaces of states and construct all relevant observables in purely

group-theoretical terms. For example, an elementary relativistic particle is described by an unitary irreducible representation of the Poincaré group [4,5]. All its states are explicitly known and may be obtained by acting with group elements on some fixed state; all observables, like energy, momentum and angular momentum (and coordinate), are constructed out of group generators. The same concerns nonrelativistic particle; the relevant group here is the (quantum mechanical) Galilei group [6].

Let us note that, in order to construct the quantum description of elementary particles, we do not have to refer to classical theory, canonical quantization, etc. However, it would be nice to find if the notion of elementary system, formulated in group-theoretical language, can be extended to classical physics in such a way that the canonical quantization of the latter yields the corresponding quantum elementary system. This question has been considered by a number of authors. The mathematical basis has been laid out by Kirillov [7,8], who developed the so-called orbit method. Souriau [9] elaborated symplectic aspects of classical and quantum physics.

Let us sketch the main points of the description of the classical system exhibiting symmetry [10,11]. The general framework for classical dynamics is provided by the Hamiltonian formalism. The space of states is a symplectic manifold—a phase space. The symplectic structure allows one to define the Poisson bracket. Once a Hamiltonian—some function on the phase space—is selected, one can write the canonical equations of motion, which determine the actual dynamics. The transformations preserving symplectic structures are called the canonical transformations. The symmetry transformations are the canonical transformations which preserve the functional form of the Hamiltonian. The set of symmetry transformations form a group, which is the symmetry group of a given dynamical system. The classical system is called elementary if the symmetry group acts transitively on phase space. It appears that in this case, the Hamiltonian formalism can be described in group-theoretical terms. To this end, let $G$ be a Lie group with Lie algebra

$$[X_\alpha, X_\beta] = ic_{\alpha\beta}{}^\gamma X_\gamma. \tag{1}$$

In the Lie algebra, $G$ acts through adjoint representation

$$Ad_g(X_\alpha) = gX_\alpha g^{-1} = D^\beta{}_\alpha(g)X_\beta. \tag{2}$$

In the dual space to the Lie algebra, there acts the contragradient representation called the coadjoint one. Let $\zeta_\alpha$ be the coordinates in the dual space; then

$$Ad_g^*(\zeta_\alpha) = D^\beta{}_\alpha(g^{-1})\zeta_\beta. \tag{3}$$

The orbit of coadjoint action (3) is called the coadjoint orbit.

In the dual space, one can define a natural Poisson structure

$$\{\zeta_\alpha, \zeta_\beta\} = c_{\alpha\beta}{}^\gamma \zeta_\gamma. \tag{4}$$

This Poisson structure is, in general, degenerate, i.e., there exist nonconstant functions having vanishing Poisson brackets with all $\zeta_\alpha$. However, the important point is that the Poisson brackets (4) can be consistently restricted to the orbits, and then they become nondegenerate; the orbits are symplectic manifolds. Moreover, if $G$ acts transitively on phase space, then, modulo some mild (i.e., fulfilled in most physical contexts) assumptions, the phase space can be identified with some coadjoint orbit of $G$. Therefore, for the elementary systems, the phase space together with its symplectic structure are described in purely group-theoretical terms. All observables, being the functions over the phase space, are expressible in terms of the coordinates $\zeta_\alpha$. One can view them as classical counterparts of the elements of enveloping algebra of the Lie algebra under consideration. What remains is the choice of the Hamiltonian, which defines the actual physical system. If $G$ describes the space-time symmetry,

one of its generators is a natural candidate for the Hamiltonian. Then, the group-theoretical description is complete: all elements of Hamiltonian formalism are expressed in terms of group notions.

The first step to classify the elementary systems with a given symmetry group is to characterize the set of coadjoint orbits. The generic orbits are the submanifolds of the dual space to Lie algebra obtained by fixing the values of all functionally independent Casimir functions (which are classical counterparts of the Casimir operators). However, there exist also nongeneric orbits of lower dimension (the extreme case being the trivial orbit). They are characterized by an additional relation, which can be found as follows. The infinitesimal action of the symmetry group on coadjoint orbit/phase space is given by the Poisson bracket with a relevant generator. Therefore, if we nullify some ideal in the Poisson algebra of functions over phase space, the resulting relations will be invariant under the coadjoint action of $G$. This is quite a convenient way of characterizing nongeneric orbits. Once the orbit is explicitly characterized, the next step is to find convenient coordinates (Darboux variables). Finally, we have to write out the Hamiltonian in terms of independent canonical variables, which completes the description.

## 3. Poincaré Symmetry

In an attempt to understand the origin of quantum mechanics of spinning particle (in particular, spin), various classical models have been proposed [12–37]. Let us sketch the description based on the ideas presented in the previous section [38].

The relativistic symmetry is described by the Poincaré group. It consists of Lorentz transformations $\Lambda(\Lambda^T \eta \Lambda = \eta, \eta = \mathrm{diag}(+---))$ and translations $a$. The composition law reads

$$(\Lambda, a) \cdot (\Lambda', a') = (\Lambda\Lambda', \Lambda a' + a) . \tag{5}$$

The relevant Lie algebra consists of Lorentz $(M_{\mu\nu})$ and translation $(P_\mu)$ generators obeying

$$[M_{\mu\nu}, P_\alpha] = i(\eta_{\nu\alpha} P_\mu - \eta_{\mu\alpha} P_\nu) , \tag{6}$$

$$[M_{\mu\nu}, M_{\alpha\beta}] = i(\eta_{\mu\beta} M_{\nu\alpha} + \eta_{\nu\alpha} M_{\mu\beta} - \eta_{\mu\alpha} M_{\nu\beta} - \eta_{\nu\beta} M_{\mu\alpha}) , \tag{7}$$

$$[P_\mu, P_\nu] = 0 . \tag{8}$$

There are two Casimir operators, mass and spin,

$$M^2 \equiv P^\mu P_\mu , \tag{9}$$

$$W^2 \equiv W^\mu W_\mu , \quad W^\mu = \frac{1}{2} \epsilon^{\mu\nu\alpha\beta} P_\nu M_{\alpha\beta} . \tag{10}$$

The relevant coordinates in the dual space to Lie algebra are $\zeta_\mu$ and $\zeta_{\mu\nu} = -\zeta_{\nu\mu}$. For physical reasons, we are interested in the case $M^2 \geqslant 0$; then, the coadjoint orbits consist of two disjoint pieces corresponding to $\zeta_0 \gtrless 0$; since $\zeta_0$, being the counterpart of time translation generator $P_0$, represents energy, we restrict ourselves to $\zeta_0 > 0$ case.

Now, the generic orbits are obtained by fixing $M^2$ and $W^2$. Therefore, they are eight-dimensional. This is exactly what we expect: there are three components of position and momentum together with two variables describing the spin of fixed length (due to fixing $W^2$). Explicitly,

$$m^2 = \zeta_\mu \zeta^\mu , \tag{11}$$

$$-m^2 s^2 = w^\mu w_\mu \,, \quad w^\mu = \frac{1}{2} \epsilon^{\mu\nu\alpha\beta} \zeta_\nu \zeta_{\alpha\beta} \,. \tag{12}$$

Note that for $m^2 = 0$, the invariants are no longer independent. Therefore, we start with $m^2 > 0$. It can be shown that the canonical point on coadjoint orbit can be chosen as [38]:

$$\underline{\zeta}_\mu = (m, \vec{0}) \,, \tag{13}$$

$$\underline{\zeta}_{0i} = -\underline{\zeta}_{i0} = 0 \,, \tag{14}$$

$$\underline{\zeta}_{ij} = s\epsilon_{3ij} \,. \tag{15}$$

Any other point of the orbit is obtained from the canonical one by a coadjoint action of the Poincaré group. By an appropriate choice of parametrization, we find that the general point of the orbit can be written as [37,38]

$$\zeta_\mu = p_\mu \,, \tag{16}$$

$$\zeta_{0i} = -p_0 x_i + \frac{\epsilon_{ilk} s_l p_k}{m + p_0} \,, \tag{17}$$

$$\zeta_{ij} = x_i p_j - x_j p_i + s_k \epsilon_{kij} \,, \tag{18}$$

while the Kirillov symplectic structure yields

$$\{x_i, p_j\} = \delta_{ij} \,, \tag{19}$$

$$\{s_i, s_j\} = \epsilon_{ijk} s_k \,, \tag{20}$$

the remaining Poisson bracket being vanishing. It is quite straightforward to quantize the resulting symplectic structure. As a result, one obtains the well-known form of generators of unitary irreducible representations of the Poincaré group describing massive particles [39].

Assume now that $m^2 = 0$. Then, both invariants vanish independently of the value $s$. Moreover, in quantum theory of massless particles, spin is no longer a dynamical variable. In fact, quantum particles are uniquely characterized by chirality, which can be viewed as the projection of spin on momentum. However, it is a fixed number, not a dynamical variable. Therefore, we expect the phase space to be six-dimensional. The corresponding orbits must be nongeneric. There is only one independent Casimir function, so, according to the prescription given above, we have to construct the relevant ideal in the Poisson algebra. Consider the following functions on phase space.

$$I_\mu(\underline{\zeta}) \equiv w_\mu - s\zeta_\mu \,. \tag{21}$$

Then

$$\{I_\mu, \zeta_\nu\} = 0 \,, \tag{22}$$

$$\{I_\mu, \zeta_{\alpha\beta}\} = g_{\mu\alpha} I_\beta - g_{\mu\beta} I_\alpha \,, \tag{23}$$

so, one can put consistently $I_\mu = 0$. $I_\mu$ are the generators of invariant ideal. The relevant coadjoint orbit is defined by the equations .

$$\zeta^\mu \zeta_\mu = 0 \,, \tag{24}$$

$$I_\nu = 0 \,. \tag{25}$$

Equation (25) is called "enslaving" condition [40,41] (cf. also [9]). Due to $\zeta^\nu I_\nu = -s\zeta^\nu \zeta_\nu = 0$, only three Equations (25) are independent. Therefore, we obtain six-dimensional orbit. Again, it is not difficult to identify some canonical point on the orbit.

$$\underline{\zeta}_\mu = (k,0,0,k) \equiv k_\mu \,, \tag{26}$$

$$\underline{\zeta}_{\mu\nu} = \begin{cases} -s & , \quad (\mu\nu) = (12) \\ s & , \quad (\mu\nu) = (21) \\ 0 & , \quad (\mu\nu) \neq (12),(21) \,. \end{cases} \tag{27}$$

By applying the coadjoint action of the Poincaré group and choosing suitable parametrization, we find [42]

$$\zeta_\mu = p_\mu \,, \tag{28}$$

$$\zeta_{0i} = -p_0 x_i \,, \tag{29}$$

$$\zeta_{ij} = x_i p_j - x_j p_i + s\frac{\epsilon_{ijk} p_k}{p_0} \,. \tag{30}$$

From Equation (30), we conclude that $s$ is the projection of total angular momentum on the momentum direction.

Equations (28)–(30), when compared with the general form of Poisson brackets (4) and Equations (6)–(8), yield [42]

$$\{x_i, x_j\} = s\frac{\epsilon_{ijk} p_k}{p_0^3} \,, \tag{31}$$

$$\{x_i, p_j\} = \delta_{ij} \,. \tag{32}$$

with the remaining brackets vanishing. Note that the coordinates do not commute any longer (cf. (31)). This is because we are considering the nongeneric orbit defined by the additional ("enslaving") condition. This situation persists on the quantum level. A nice argument can be given [43,44] that it is not possible to define standard, i.e., commuting, coordinates for massless irreducible representation; only for reducible representation describing the helicities $S = \pm\frac{1}{2}$ such a coordinate operator exists [44].

Due to the more complicated form of Poisson brackets, the canonical quantization procedure is nontrivial. The coordinates cannot be represented by derivatives with respect to momentum; instead, a covariant derivative in the field of monopole must be used.

The approach described above provides a systematic way of studying symmetries on the level of classical Hamiltonian formalism. Phase space and dynamical observables are constructed in terms of group-theoretical notions and put on firm ground. In particular, one can study the transformation

properties of various observables under the action of the symmetry group. This concerns, for example, the coordinate variable. It appears that it has "exotic" transformation properties. Consider, as an example, a massless particle with helicity $s$. The conserved generator of boosts reads (cf. Equation (29))

$$\zeta_{0i}(t) = p_0 x_i - p_i t .\tag{33}$$

By virtue of Equation (31), one finds [40–42,45,46]

$$\delta \vec{x} = \{\vec{x}, \delta v_i \zeta_{0i}(t)\} = -\delta \vec{v} t + \dot{\vec{x}}(\delta \vec{v} \cdot \vec{x}) + s \frac{\delta \vec{v} \times \vec{p}}{p_0^2} .\tag{34}$$

The first term on the right-hand side corresponds to the standard Lorentz transformation, while the second appears due to the fact that in the Hamiltonian formalism, time is kept fixed, so one has to recompute everything back to the initial time. The last term, which is helicity dependent, represents the so-called "side jump". The latter leads to the kinematical effect playing a role in impurity scattering caused by spin-orbit interaction [47] and relativistic Hall effect of light [48–54].

## 4. The Conformal Group

In four-dimensional space-time, the conformal group provides a fifteen-dimensional extension of the Poincaré group by scaling and special conformal transformations. It describes the approximate symmetry at energies large as compared to all dimensionful parameters. It would be interesting to provide the Hamiltonian description of all elementary conformally invariant dynamical systems.

The dual space to conformal Lie algebra is parametrized by $\zeta_\mu$, $\zeta_{\mu\nu}$ (Poincaré algebra), $\eta$ (dilatations) and $\eta_\mu$ (special conformal transformations). Apart from the Poisson brackets following from commutation rules (6)–(8), we have the additional ones

$$\{\eta, \zeta_\mu\} = \zeta_\mu ,\tag{35}$$

$$\{\eta, \zeta_{\mu\nu}\} = 0 ,\tag{36}$$

$$\{\eta, \eta_\mu\} = -\eta_\mu ,\tag{37}$$

$$\{\zeta_{\mu\nu}, \eta_\rho\} = g_{\nu\rho} \eta_\mu - g_{\mu\rho} \eta_\nu ,\tag{38}$$

$$\{\eta_\mu, \zeta_\nu\} = 2(\zeta_{\mu\nu} - g_{\mu\nu}\eta) ,\tag{39}$$

$$\{\eta_\mu, \eta_\nu\} = 0 .\tag{40}$$

In order to classify the conformally invariant Hamiltonian system, we have to find all coadjoint orbits. It is not completely straightforward. The convenient way to do this is to use twistor formalism [55]. There are coadjoint orbits of dimensions 12 (generic), 10, 8 and 6. The latter case is particularly interesting: the Poincaré symmetry of quantum massless particles can be extended to the conformal one [56,57].

Therefore, we expect that the same holds true classically. In order to characterize six-dimensional orbits, one has to find nine independent generators of the relevant ideal. We already have four generators: $\zeta_\mu \zeta^\mu$ and $I_\mu$ (cf. Equation (21)). The remaining five can be chosen as [58]

$$J \equiv \eta + \frac{\zeta_k \zeta_{0k}}{\zeta_0} , \tag{41}$$

$$J_0 \equiv \eta_0 + \frac{\zeta_{0k} \zeta_{0k}}{\zeta_0} + \frac{\lambda^2}{\zeta_0} , \tag{42}$$

$$J_i \equiv \eta_i + \frac{\zeta_i \zeta_{0k} \zeta_{0k}}{\zeta_0^2} - 2 \frac{\zeta_{0i} \zeta_k \zeta_{0k}}{\zeta_0^2} - 2s\epsilon_{ikl} \frac{\zeta_{0k} \zeta_l}{\zeta_0^2} - s^2 \frac{\zeta_i}{\zeta_0^2} . \tag{43}$$

Equations (41)–(43) merely imply that the generators of dilatations and special conformal transformations are functions on the phase space of Poincaré covariant massless particles

$$D = +p_k x_k , \tag{44}$$

$$K_i = -p_i x_k x_k + 2x_i x_k p_k - 2s\epsilon_{ikl} \frac{x_k p_l}{p_0} + \frac{s^2 p_i}{p_0^2} . \tag{45}$$

We conclude that, on the classical Hamiltonian level, the Poincaré symmetry of massless particles of arbitrary helicity may be extended to the conformal one. One can show [58] that the canonical quantization of the resulting structure can be performed in a more or less straightforward way, leading to the unitary irreducible representations of the conformal group belonging to the Mack list [57].

The dynamical systems corresponding to the orbits of higher dimensions will be described elsewhere [59].

## 5. Conclusions

Canonical quantization is the textbook method of constructing the quantum dynamics. One starts with some classical dynamical system defined within the Hamiltonian formalism and applies the canonical quantization procedure consisting in replacing the Poisson brackets by commutators (divided by $i\hbar$). With a little bit of luck, a consistent mathematical structure is obtained, which defines the quantum counterpart of the classical system we have started with.

However, the quantum dynamics should be the primary concept with the classical one emerging in the appropriate limit. Therefore, the question arises as to how to construct the quantum theory without referring to classical notions. One (the only?) way to do this is to refer to the symmetry arguments. According to the basic principles of quantum mechanics, the symmetry transformations are represented by unitary operators acting in the Hilbert space of states. Given a symmetry group $G$, all allowed spaces of states of the physical system can be classified and explicitly described, provided that the (projective) unitary representations of $G$ are known. Furthermore, by identifying the elementary systems with irreducible representations one concludes that, in this case, all observables can be, at least in principle, constructed from the elements of the Lie algebra of $G$.

When such a construction is performed, one may consider (at least on a formal level) the limit $\hbar \to 0$, yielding some classical dynamics, and pose the question as to whether the former can be recovered by applying the canonical quantization to the latter. The answer to this question is affirmative in the case of relativistic space-time symmetries. We have sketched above the main steps of the relevant construction. The starting point is the notion of the elementary Hamiltonian system: it is the one exhibiting the symmetry which acts transitively on the phase space. Then, the candidates for the

admissible phase spaces are provided by the coadjoint orbits of the symmetry group. It appears that both generic and nongeneric orbits should be considered; for example, the massless particles are described by nongeneric orbits of the Poincaré group. Once the orbit is selected, the explicit description of the resulting classical Hamiltonian system is obtained (one should keep in mind that for space-time symmetries, the Hamiltonian belongs to the Lie algebra of symmetry group).

Thus far, the above program has been completed for the Poincaré symmetry. The next step is to extend it to the conformal group. We have already shown [58] that the nongeneric six-dimensional coadjoint orbits of the conformal group describe massless particles; more precisely, in this case, the Poincaré symmetry can be extended to the conformal one. Canonical quantization (some care concerning the ordering problem must be exercised) of the generators of conformal group yields its unitary representation acting in the space of states of massless particles with fixed helicity; this result agrees with the one obtained by Mack [57]. It remains to consider the nongeneric orbits which are eight- and ten-dimensional as well as generic, twelve-dimensional ones. They should provide the classical counterparts of the remaining representations classified by Mack.

**Author Contributions:** Both authors contributed equally to this work. All authors have read and agreed to the published version of the manuscript.

**Funding:** This research received no external funding.

**Acknowledgments:** We are grateful to Joasia Gonera, Krzysztof Andrzejewski and Cezary Gonera for numerous, pleasant, and enlightening discussions.

**Conflicts of Interest:** The authors declare no conflict of interest.

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
