# Peer review of "Relativistic Symmetries and Hamiltonian Formalism"

_symmetry, doi:10.3390/sym12111810_

Round 1

Reviewer 1 Report

This paper is interesting,  The authors like to interpret the Poincare symmetry, particularly that of massless particles, within the framework of the conformal group. However, they do not interpret the Poincare symmetry, especially that of massless particles, correctly. The authors may study Wigner's last paper on this subject:

Kim and Wigner, Space-time symmetries of relativistic particles, published in the Journal of Math. Phys. [31] 55-60 (1990).

Author Response

Dear reviewer,

thank you for your report. However, we see no contradiction between our results and those contained in the paper you quoted. Our expressions for generators of Poincare transformations (eqs.(16)-(18) and (28)-(30)) obey correct (Poisson) commutation rules. After quantization they yield correct generators of unitary irreps of Poincare group.

Kind regards, The authors

Reviewer 2 Report

It is a nice brief review of the orbifold formalism applications to relativistic symmetries. The paper is based on earlier publications of the authors. But the combined unified presentation of the results looks useful and it should be interesting for readers.

There is one important remark. The absence of a Conclusions section makes the paper unfinished. There is just one phrase claiming that orbits of higher dimesions will be considered elsewhere. I would suggest to add some more concluding remarks with an outlook.

As concerns the presentation, the text is written clearly in a good language. The sufficient amount of formulae is given. It might be worth to put punctuation marks in equations (commas and dots are systematically omitted there).

I suggest publishing the paper upon the author's reply to the above remarks.

Author Response

Dear Reviewer,

thank you for your favorable report. We have added new section "Conclusions" and punctation marks in equations. 

Kind regards, The authors

Reviewer 3 Report

The manuscript offers a useful compact review on the application of Lie groups

to relativistic dynamical systems. 

Two editorial comments: In line (26), one expects references related to " anomaly related phenomena....  In line 110, the metric tensor \eta = diag(..) should be specified.

Author Response

Dear Reviewer,

thank you for favorable report. We specified eta (above eq.(5)) and added two references ( [1] and [2]) concerning anomaly related phenomena (chiral magnetic effect etc.).

Kind regards, The authors

Reviewer 4 Report

l 10) 'the coadjoint' l 15-16) 'us and our colleagues in collaboration with Yves Brihaye' --> 'our colleagues in collaboration with Yves Brihaye and us' l 17) 'the basic' l 18) ', which' l 19) 'theory, ' l 21) 'a straightforward' l 23) 'years, '; 'explore' -> 'exploring' l 24) 'to much extent' -> ', to a great extent, ' l 27) 'a deeper' l 29) 'in collaboration' l 33) 'physics, ' l 34) 'data, ' l 37) ', which' l 38) ', and' l 40) 'a guiding'; 'leads often' -> 'often leads' l 42) 'the physical' l 43) 'by choice of'; 'theory, ' l 48) ', which' l 49) 'the symmetry'; 'the elementary' l 52) 'an elementary' l 53) 'the Poincare' l 56) 'coordinate), ' l 60) 'particles, ' l 61) ', etc.' l 66) ', who' line number missing 69) 'the classical' line number missing l 73) 'to define' -> 'defining' line number missing l 74) ', one' line number missing l 75) ', which' line number missing l 76) 'structures' line number missing l 79) ', which' line number missing l 82) 'case, ' line number missing l 83) 'end, ' line number missing l 85) 'algebra, G' line number missing l 87) 'algebra, there' line number missing l 70) 'dual space, ' l 70) ' i.e.,' l 73) ', and' l 75) 'space then,' -> 'space, then'; 'i.e.,' l 76) 'assumptions, ' l 77) ', the'  l 83) ', which'  l 93) 'an additional' l 94) ', which'  l 96) 'a relevant' l 97) 'phase space,' l 98) 'a convenient ' l 100) 'characterized,' l 102) ', which' l 104) 'an attempt' l 105) 'spin),' above eq5) 'the Poincare' below eq 10) 'reasons,' ; 'then, the coadjoint';  above eq 11) 'eight-dimensional'; 'describing the spin' below eq 12) '0, the' below eq 15) 'the Poincare'; 'parametrization, we' l 109) 'result, ' l 111) 'the Poincare' above eq 21) 'Moreover, the quantum theory of massless particle spin'; 'by chirality, which can '; 'six-dimensional.'; 'Casimir function, so according' above eq 26) 'six-dimensional.' above eq 28) 'the Poincare'; 'parametrization,'   l 115) 'the quantum' l 116) 'the standard, i.e., commuting coordinates' l 120) 'Poisson brackets,' above eq 33) 'the symmetry group'; 'Consider, as an example, a massless' l 124) 'right-hand ' l 126) 'formalism,'; 'fixed, so' l 127) 'term, which' l 128) 'dependent,' l 129) 'the kinematical '; 'playing a role in' l 132) 'In four-dimensional space-time,'; 'a fifteen-dimensional' l 133) 'of the Poincare' above eq 35) 'Apart from the'; '), we have ' l 138) 'system,' above eq 40) 'six-dimensional orbits' l 144) ' level, ' l 147) 'in a more' l 148) 'the conformal' l 156) 'nice and' -> 'pleasant, and' 

Author Response

Dear Reviewer,

thank you for your detailed remarks. We corrected the manuscript according to your suggestions.

Kind regards, The authors

Reviewer 5 Report

In this manuscript the authors re-examine the reltivistic symmetries of the Poincare group, a subject of historial interest, but largely known at present, having being well explained not only in the literature but also in textbooks.

The claimed origininality of the present paper is ithat in this manuscript the author present the problem in a Hamiltonian framework. In my opinion this not add enough substance to mek the paper interesting.

Finally, the authors claim that the interest of their work is not only academic but has potential applications. I do not see any of those applications in the text.

Summarizing, in my opinion the content of the manuscipt is weel known and add very little to the present knowledge of the subject , which is a well known topic in relativity. As a consequence I recommned rejection of the manuscript.  

Author Response

Dear Reviewer,

thank you for your report. Let us say the following:

  1. Our paper is the contribution to the volume celebrating Yves Brihaye. Therefore, we reviewed the topics we were working on in collaboration with him. That is why we didn't describe the applications. However, we quote, as an example, the paper [39] (this is new number because we have added two references) where the issue of Lorentz invariance in chiral kinetic theory is studied. The problem is decribed there in some detail.
  2. In last five years we have published four papers on the problems discussed in the paper under consideration (in Phys. Lett. B, JHEP and Ann. Phys.). Moreover, other authors also worked on the related problems; see [39], [35], [36], [47] and other. As for the issues explained already in textbooks it is pretty good.
  3. Kind regards, The authors

Round 2

Reviewer 1 Report

The problem of this paper is that the authors do not understand the Poincare symmetries of massive and massless particles.  The basic Poincare symmetry is like the three-dimensional rotation group, and the symmetry of massless particles is like the two-dimensional Euclidian group leading to helicity and gauge transformations.  The transition from the massive case to the massless case requires a procedure of group contractions.

These aspects of the Poincare symmetry are well known.  Since the authors are able to reflects these points, I am not able to recommend this paper for publication.